# Calibrating agent-based models to tumor images using representation learning

**Colin G. Cess**[1], **Stacey D. Finley** [1,2,3]*

**1** Alfred E. Mann Department of Biomedical Engineering, University of Southern California, Los Angeles, California, United States of America, **2** Department of Quantitative and Computational Biology, University of Southern California, Los Angeles, California, United States of America, **3** Mork Family Department of Chemical Engineering and Materials Science, University of Southern California, Los Angeles, California, United States of America

* sfinley@usc.edu

**Data Availability Statement:** Code used for formatting input data (tumor images) and ABM simulations, training the neural network to represent data and simulations as low dimensional points, and calculating the distance between data

## Abstract

Agent-based models (ABMs) have enabled great advances in the study of tumor development and therapeutic response, allowing researchers to explore the spatiotemporal evolution of the tumor and its microenvironment. However, these models face serious drawbacks in the realm of parameterization – ABM parameters are typically set individually based on various data and literature sources, rather than through a rigorous parameter estimation approach. While ABMs can be fit to simple time-course data (such as tumor volume), that type of data loses the spatial information that is a defining feature of ABMs. While tumor images provide spatial information, it is exceedingly difficult to compare tumor images to ABM simulations beyond a qualitative visual comparison. Without a quantitative method of comparing the similarity of tumor images to ABM simulations, a rigorous parameter fitting is not possible. Here, we present a novel approach that applies neural networks to represent both tumor images and ABM simulations as low dimensional points, with the distance between points acting as a quantitative measure of difference between the two. This enables a quantitative comparison of tumor images and ABM simulations, where the distance between simulated and experimental images can be minimized using standard parameter-fitting algorithms. Here, we describe this method and present two examples to demonstrate the application of the approach to estimate parameters for two distinct ABMs. Overall, we provide a novel method to robustly estimate ABM parameters.

## Author summary

Parameter estimation is a key step in computational model development, and accurate parameters are required to produce robust model predictions. Agent-based models (ABMs) are commonly used to simulate tumor growth; however, these models are exceedingly difficult to fit to experimental or clinical imaging data due to the complex spatial relationships of various cell types. Currently, simple comparison metrics extracted from tumor images and ABM simulations are used to qualitatively assess the model fit. In this work, we present a novel method for comparing spatial ABM simulations to tumor images

and model simulations is available on a GitHub repository at https://github.com/FinleyLabUSC/Representation-learning-for-ABM-parameter-estimation. In addition, we provide the code for the ABMs.

**Funding:** SDF has received support from the USC Center for Computational Modeling of Cancer. SDF received a Research on Engineering Medicine for Cancer grant from the USC Ming Hsieh Institute. The funders had no role in study design, data collection and analysis, decision to publish, or preparation of the manuscript.

**Competing interests:** The authors have no competing interests.

as a single quantitative value that measures how different the two are and can then be used as the objective function for a parameter estimation algorithm. Our approach uses representation learning, where a neural network is used to project an input into low-dimensional space. This method can be used to aid researchers in developing and fitting tumor ABMs based on actual patient data.

## Introduction

Agent-based models (ABMs) of cellular systems have been used to explore various facets of physiology and disease [1,2]. Specifically in the field of oncology, ABMs have been used to explore the many different ways in which the tumor is influenced by the microenvironment, such as through hypoxia, angiogenesis, invasion, and interactions with the immune system [3–5]. While ABMs are powerful tools that can simulate many different tumor properties and predict emergent behaviors that occur on the spatial level, they are limited in the realm of parameterization. In general, fitting spatial models of cell populations to spatial data is a difficult task, where specific features of the spatial state have to be extracted in order to perform comparisons [6]. For ABMs, key parameters, such as proliferation rates, cell lifespans, and migration rates, can often be experimentally measured outside of the scope of the modeled system and are commonly found in the literature. However, this sometimes leaves specific parameters with unknown values. Many of these parameters are often phenotypic summaries of complex biological mechanisms, reducing phenomena such as the secretion of cytokines and subsequent activation of intracellular signaling pathways to simple interactions governed by a single parameter [7–11]. Therefore, an ABM needs to be compared to data in order to properly estimate the unknown parameter values.

The spatially-resolved nature of ABMs makes parameter estimation more difficult than in equation-based models. For an equation-based model, while parameter estimation is not a trivial task, it is straightforward, requiring a quantitative comparison of the model outputs and experimental data. In most cases, the experimental data are represented by a single, continuous value, which is the same format as the model output. This makes it easy to calculate the distance between experimental data and model simulation [12]. One can then use parameter estimation algorithms that minimize the distance between the two [13–16]. For ABMs, however, this is more complicated. While a comparison to quantitative data, such as tumor volume over time, is possible, this type of data loses the spatial aspect that is a defining feature of an ABM. Therefore, in order to better estimate unknown parameters, a comparison to tumor images, which provide spatial information, would be beneficial.

Comparison to tumor images poses a challenge in that it is very difficult to compare imaging data to model simulations. For ABMs of tumors, simple qualitative comparisons are often performed to determine how well the model represents the spatial biology. In some cases, simple metrics are extracted and used for comparison [17–20]. However, this fails to capture the full extent of the spatial properties being modeled. Furthermore, selecting comparison metrics introduces bias to the parameter estimation, becomes unwieldy as model complexity increases, and can potentially miss complex spatial interactions that are hard for the researcher to detect.

To this end, we endeavored to develop a method for quantitatively comparing experimental tumor images to ABM simulations as a scalar metric. We have previously developed an approach for learning low-dimensional representations of model simulations using neural networks, which allows us to compare complex model simulations in terms of a single distance metric [21]. Here, we extend that approach to comparing ABM simulations to tumor images.

The closer the projected points for two inputs to the neural network are, the more similar the inputs are. We are able to use this distance between a tumor image and ABM simulation as the objective function for parameter estimation. Neural networks are widely used in various forms of image analysis and have previously been shown to work better than methods such as PCA for image dimensionality reduction [22].

Here, we display how we process both fluorescent images and the spatial output from ABM simulations into a similar format so that we can compare them with our neural network method. We detail how we train a neural network to learn low-dimensional representations of this type of model output and how this can be used as an objective function for parameter estimation. We present two examples to demonstrate the application of this method. In the first test case, we fit a model to its own simulations to display method functionality in a controlled setting. In the second example, we fit an ABM to a fluorescence image from an *in vitro* device that recapitulates a hypoxic tumor microenvironment, displaying how this method works on real data and enables fitting an ABM across spatial scales.

## Methods

This fitting approach is based on our previously developed method for comparing complex model simulations [21]. In brief, that method uses representation learning to train a neural network on Monte Carlo simulations of a model to project model simulations into low-dimensional space. As a result, the difference between simulations is given by the distance in low-dimensional space. The method enables a holistic comparison of model simulations without the need to manually calculate comparison metrics. Here, we train a neural network on model-generated data from an ABM and use it to compare model simulations to a tumor image. This method is very similar to our previous one, with the main distinction here being the pre-processing of simulations and tumor images so that they are in a comparable format. After projecting the processed model simulation and tumor image into low-dimensional space, we can use the distance between projected points as an objective function. This allows us to use a parameter estimation algorithm to minimize the distance between projections, thus fitting the ABM to the tumor image. We display a schematic of this process in **Fig 1**.

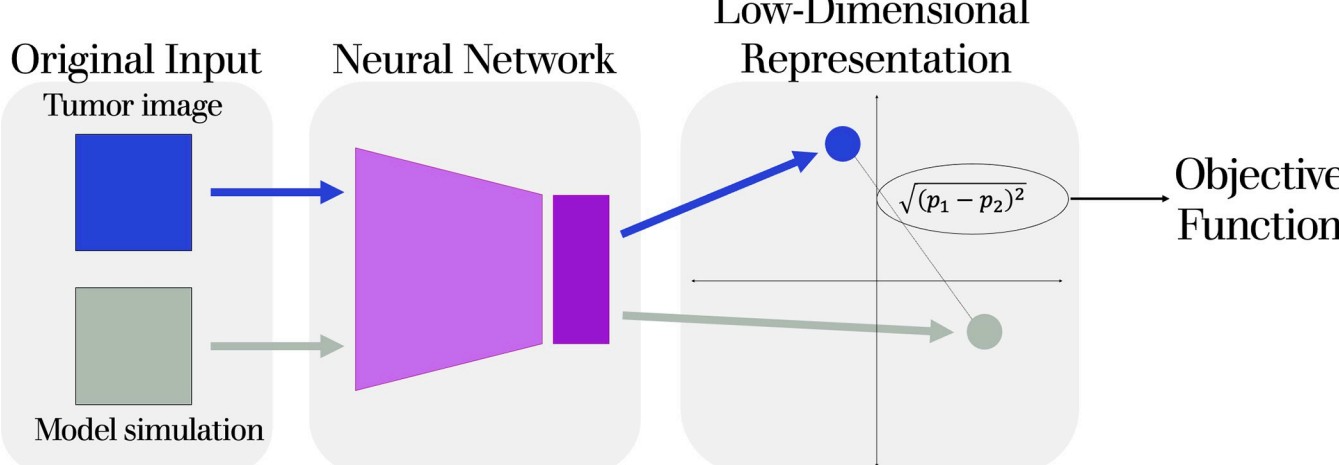

**Fig 1. Schematic displaying how two inputs, a tumor image and the spatial output from an ABM simulation, are inputted into the same neural network and projected to low-dimensional space.** The Euclidean distance between these projected points is used as the objective function to be minimized with a parameter estimation algorithm.

## Data processing

We start by describing the data-processing step, as it is a key piece of this study. We note that the processing we describe here is specific to our use-case with ABMs and tumor images. However, using a different data-processing step would allow our method to be used with other types of models and imaging data. Additionally, this is only one potential way to process images and simulations, and other methods may be able to yield similar effects. The exact implementation of data-processing is based on the model framework and available data. In their original formats, ABM simulations and fluorescence images have little in common. ABMs output a list of cells with their coordinates and properties, while a fluorescence image is a picture with several color-channels, one for each fluorescence stain. The first step of processing is to extract the cell coordinates from the fluorescence image, which can be done using readily available software. Here, we use the *Analyze Particles* function in ImageJ [23]. Based on the fluorescence stains, we now have converted the image to a list of cells with their coordinates and properties, the same as the ABM simulations. After this, the cell lists for the simulations are pruned to only contain the same cell types and properties as that from the image.

The second step of data-processing involves converting the cell lists back into an image format, which we term the "simplified images." This is done to both the cell lists from tumor images and from model simulations. By converting the cell lists into images, we can make use of convolutional neural networks, which are used extensively for image analysis. Additionally, by converting from cell lists, we generate images that are qualitatively comparable and in the same format. We form the simplified images by discretizing cell coordinates from their continuous values to a grid, where one grid space is the size of one cell diameter, akin to the format of an on-lattice ABM. We have one grid per cell type or property, yielding a *three*-dimensional matrix for each image or simulation, with the first two dimensions corresponding to the *x,y*-dimensions and the remaining dimension for cell types and properties. Each cell type or property is represented as a grid that corresponds to a color-channel, and each site within the grid corresponds to a pixel in the simplified image. Grids for cell types take on values of zero or one, indicating whether or not that cell type is present. Grids for other properties, such as ligand expression, have continuous values, and we scale these so that each grid has a maximum value of one, since values in the model do not directly equal that of fluorescence intensity.

The final step resizes the simplified images to a smaller, uniform size. This serves two purposes. The first is so that it produces the inputs that can be used in convolutional neural networks, which require images of the same size. The second is that it aggregates discrete cell locations into regions of cell density, allowing us to compare across spatial scales. This is important since tumor images are generally of a larger size than is computationally feasible to simulate with an ABM. Before resizing, we crop the simplified images to be bound to the dimensions of the tumor. Then, to resize the images, we use the *resize* function from the OpenCV Python library, using area interpolation [24]. We then rescale each grid such the values for all grid sites are between zero and one. A schematic of this final processing step is shown in **Fig 2A**, with an example from our first test model (described in a later section) shown in **Fig 2B**, where it is separated into four grids, with three for cell locations and one for ligand expression. We note that this step brings with it some considerations for parameter estimation, which we cover in the Discussion.

Overall, this processing step serves to convert both model simulations and tumor images into coarse-grained images showing cell densities. While this does lose some of the more detailed spatial features, it is necessary in order to have both simulations and images in a comparable format.

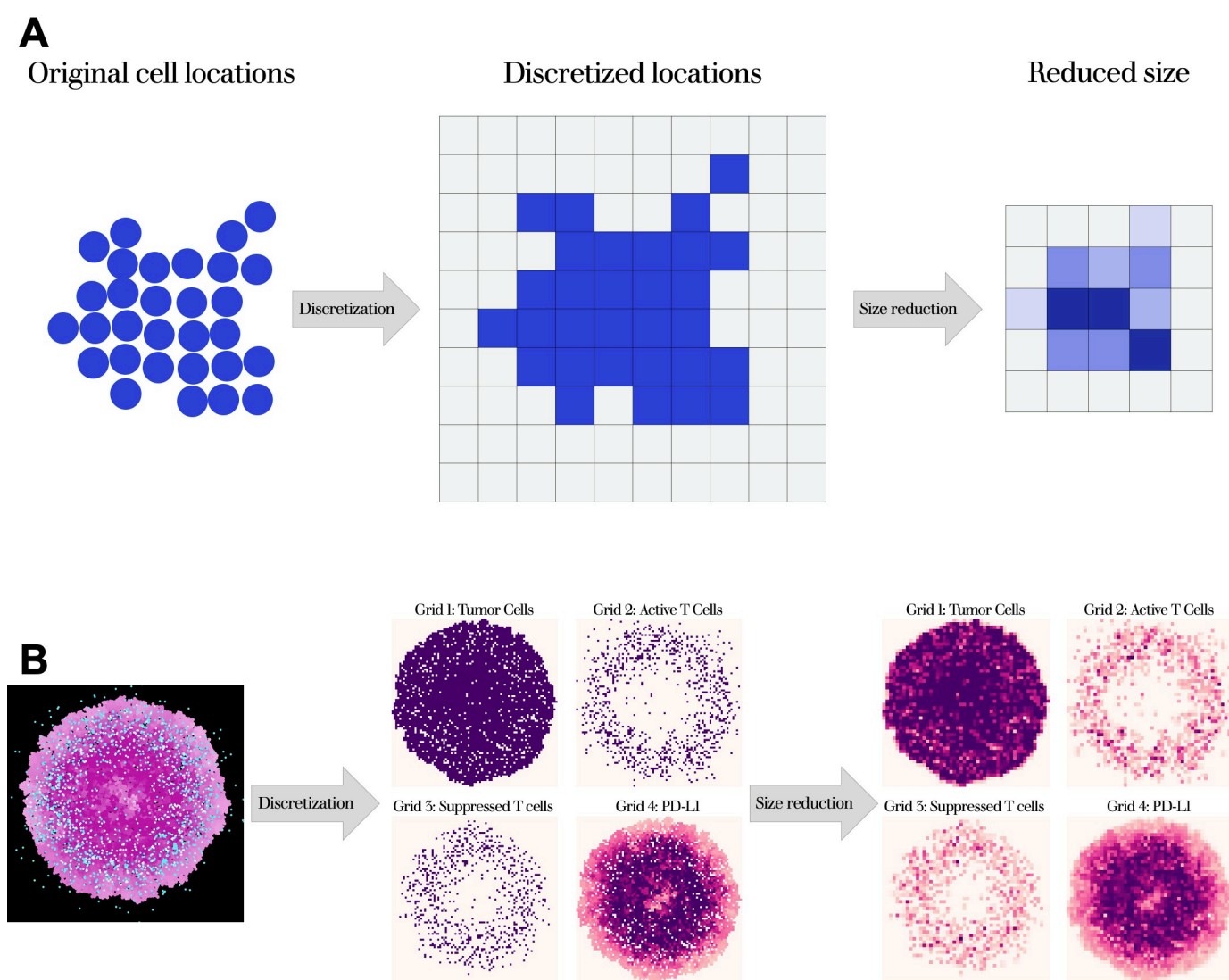

**Fig 2.** Data-processing schematic (A) and example simulation (B). From the continuous spatial layout, cells are separated by property and discretized to a series of grids. The grids are then reduced to a smaller size, converting discrete locations to densities. In (B, left), pink dots are tumor cells (darker = higher PD-L1), teal dots are active T cells, and white are suppressed T cells. In (B, right), cell densities range from light pink to purple. For ease of visualization, we show the same size figures in the center and right panels, though we note there are fewer grid spaces in the right panel, compared to center.

## Generation of training data

Following establishment of an image processing protocol, the next step of our method is to generate the model simulations (to which the processing protocol is applied). These simulations are needed to train the neural network. We randomly sample each parameter to be fitted over the widest range of potential values, generating 10,000 Monte Carlo simulations. Because agent-based models are stochastic, we perform simulation replicates to capture differences in model behavior. Once simulations are completed, we process them using the above method. To simplify method development, here we only vary the model parameters. However, especially in cases where the initial conditions are unknown, it could be useful to also vary the initial conditions.

## Training of the neural network

We then use our processed dataset to learn representations of the model output using the approach outlined in SimCLR, which we applied in our previous work. This approach requires augmentations of the training inputs, which we perform by randomly mirroring and rotating each simplified image. The goal of this training approach is to move the representations of two augmentations of an input closer together in projected space than they are to augmentations of other inputs. Full details of this method can be found in previous works [21,25].

We train the neural network to model-generated data. The reason for using simulated data instead of training to biological images is that there are generally very few images obtained in a specific experimental study. In comparison, a model can generate a large amount of training data relatively quickly. This means that the neural network can better learn to generate representations of a broader range of model behaviors, thus improving the accuracy of the comparisons to image data.

Here, we project to two-dimensional space. It is important to keep the number of projected dimensions low, as Euclidean distance becomes less accurate in high dimensions. We recommend using the lowest dimension that yields meaningful results. We train an ensemble of 50 neural networks on 10,000 simulations and average their predicted distances when applying them to parameter fitting.

## Initial estimation of parameter ranges

Now that we have an ensemble of neural networks that are trained to create representations of our model, we can use it to estimate the model parameters. We first determine the general region of parameter space that the tumor image sits in, to avoid having to search the entire parameter space, thus reducing the computational time needed for fitting. To do this, we use the neural network to project both the processed tumor image and the training simulations into low-dimensional space. We then take the *n* closest simulations to the image and examine their parameter values, using the minimum and maximum parameter values as the lower and upper bounds for parameter estimation. A schematic of this is shown in **Fig 3**.

## Example models

Now that we have described our approach, we will detail the models that we use to test it. We note that we are using these models simply to test our method, and not to produce biological insight regarding tumor growth. In Example 1, we fit an ABM to its own simulation results. This model simulates the interactions between T cells and tumor cells. This model is a center-based model, meaning that each cell is represented by a point and a radius [26]. Tumor cells proliferate at a set rate and gain expression of the immune checkpoint inhibitor PD-L1 at a set rate in the presence of T cells [20]. The effect of PD-L1 is represented as a probability of the tumor cells suppressing a nearby T cell at each timestep. T cells are recruited around the tumor and migrate towards the tumor center up to a specified distance. T cells kill nearby tumor cells at a set probability. T cells suppressed by PD-L1 no longer migrate or kill tumor cells. We fit four parameters, based on how strongly they influence model simulations: T cell killing probability, T cell infiltration distance, maximum PD-L1 probability, and rate of PD-L1 increase.

The model for Example 2 was designed to be fitted to a fluorescence image taken from Ando *et al*. (Fig 5A of [27]), which stained for living and dead cells [27]. The authors of that study examine CAR T cell interactions with tumors via a tumor-on-a-chip model. The tumor is represented by a thin region of tumor cells, and T cells enter from the edge and migrate into the tumor. Their system also accounts for hypoxia in the tumor center. To model this, we

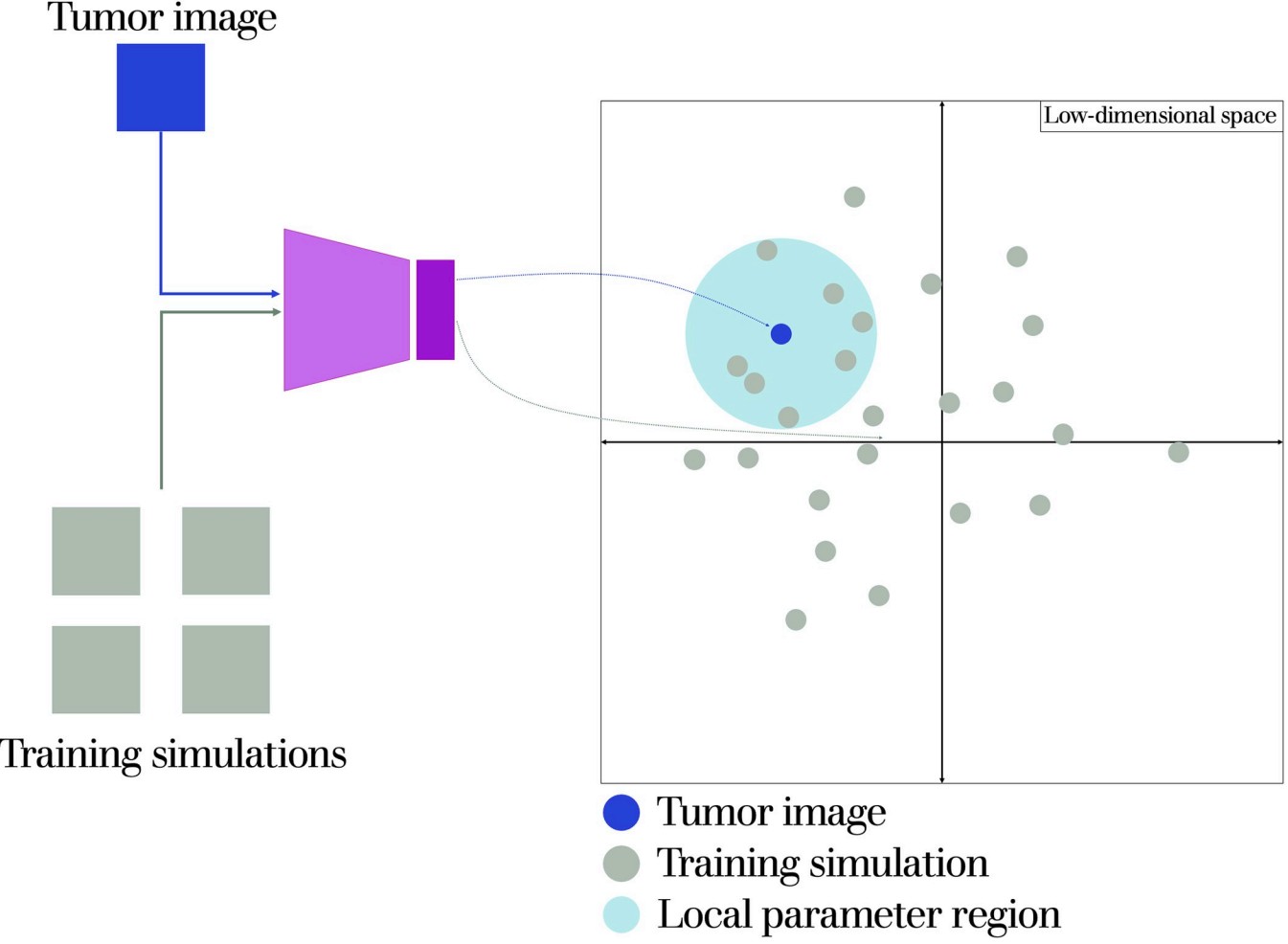

**Fig 3. Schematic displaying the initial estimation of parameter ranges.** The tumor image and the Monte Carlo simulations used for training the neural network are projected into low-dimensional space. Parameter values for the n closest simulations to the tumor image are compared and used to set the upper and lower bounds for parameter estimation.

make two modifications to the previously described model used in Example 1. The first is that we add a hypoxic region in the tumor center, where any cells within the region have an increased probability of spontaneous death. The second is that we do not remove dead tumor cells from the environment. We note that these simple extensions do not account for all of the complex biological mechanisms occurring in the *in vitro* system. However, the purpose of this paper is to display how we can create a quantitative metric that can be used in an objective function for fitting ABMs to images. Thus, we kept this model as simple as possible. Here, we fit the following five parameters, which are context-dependent and should be fit to data: T cell killing probability, T cell infiltration distance, basal tumor cell death probability, size of the hypoxic area, and probability of dying from hypoxia.

## Results

Here, we show fitting examples for the two test models, displaying the functionality of our approach. We only show the results of fitting and do not infer any biological implications of the results, as the focus of this study is to display the use of representation learning for

specifying objective functions. In addition, we keep the fitting process simple overall, as our focus is on the objective function and not parameter estimation as a whole.

### Example 1: Fitting to model-generated data

With our first test model, we produced a base simulation by manually setting the parameters that would be fitted. The purpose of this is to test our method in a controlled scenario, where we know what the fitted parameters should be. Thus, a strong demonstration of the method will produce fitted parameters that are close to those of the base simulation.

We fit four parameters that strongly influence the number and spatial distribution of cells in the ABM using the approach described in the Methods. The base simulation was projected to low-dimensional space and the nearest 100 training simulations were used to set the upper and lower bounds for parameter estimation. This number can be adjusted based on the model being fit. Parameter estimation was then performed using a genetic algorithm (GA) consisting of 300 individuals, with the objective function being the distance in low-dimensional space between the base simulation and the fitting simulations. A genetic algorithm was chosen simply because of easy of implementation on the computing cluster. However, other estimation algorithms can be used with the objective function produce in our method. We only perform one simulation replicate for each parameter set in the GA, in order to improve computational time. The GA converges to similar parameter sets which produce the best fit, thus creating simulation replicates that account for the inherent stochasticity of ABMs.

The results of this fitting are shown in **Fig 4** and **Table 1**. We see that the average and best fits quickly level off after only a small number of fitting steps (**Fig 4A**). In **Fig 4B**, we display the end-point of the base simulation together with the best fit, providing a visual confirmation that the fitted model captures the baseline simulation. The best-fit parameters are very close to

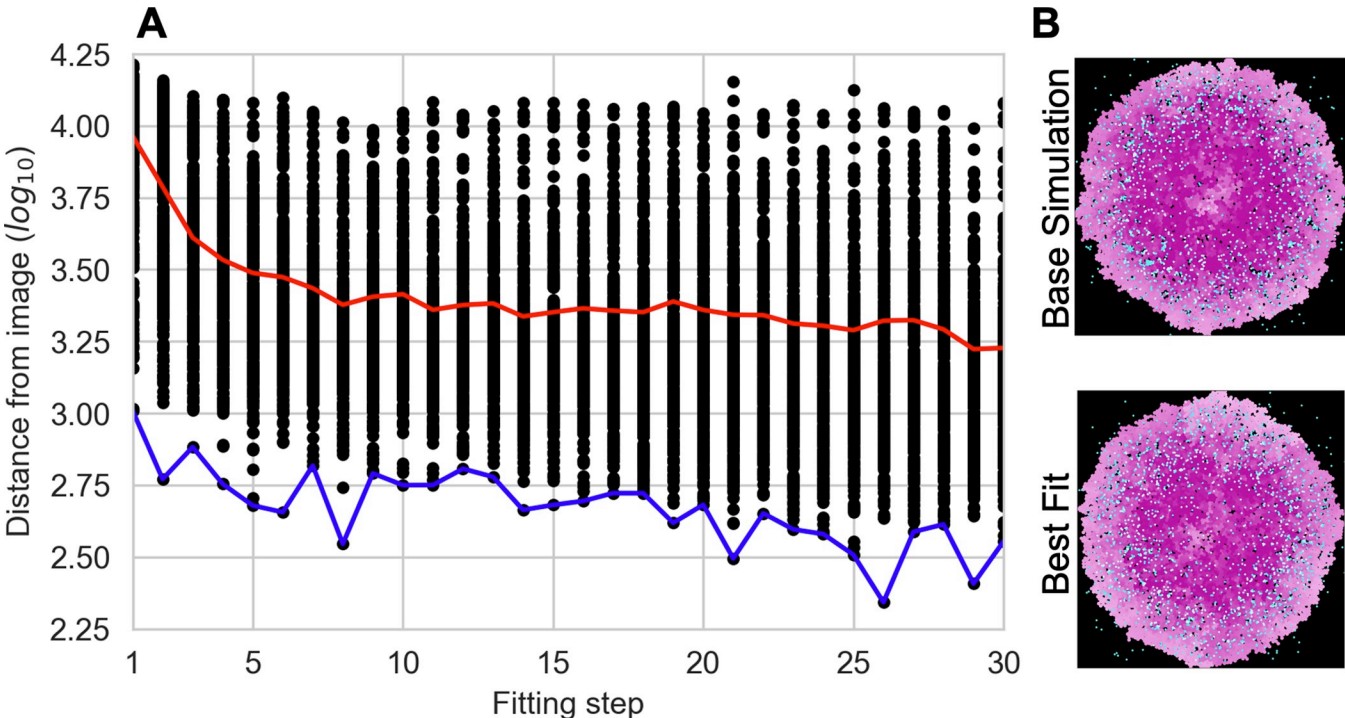

**Fig 4. Fitting results for Example 1.** (A) Fitting with a GA. Black dots–individuals in the GA. Red line–average fit. Blue line–best fit. (B) Visual comparison of the spatial layouts of the base simulation (top) and the best-fit simulation (bottom).

**Table 1. Top 10 best fits from fitting an ABM to model-generated data for Example 1.**

|  | T cell killing probability | T cell infiltration | Maximum PD-L1 | Rate of PD-L1 increase |
|---|---|---|---|---|
| **Nominal Parameters** | **0.02** | **0.8** | **0.01** | **5x10⁻⁵** |
| **Best Fit** | 0.0186 | 0.853 | 0.0102 | $4.57 \times 10^{-5}$ |
| **Maximum** | 0.0199 | 0.875 | 0.0106 | $5.21 \times 10^{-5}$ |
| **Minimum** | 0.0186 | 0.787 | 0.0068 | $3.62 \times 10^{-5}$ |
| **Mean** | 0.0192 | 0.827 | 0.0097 | $4.52 \times 10^{-5}$ |
| **Coefficient of Variation** | 0.0198 | 0.0438 | 0.1062 | 0.0810 |

the nominal parameters (**Table 1**), and the maximum, mean, and minimum for the top 10 fits are fairly constrained around the nominal parameters. These results show that learned representations have potential for use as objective functions to compare ABM simulations to images.

## Example 2: Fitting to in vitro fluorescence image

Next, we test our approach using real-world data. Here, we apply the same method as in the previous section, with the exception of using 400 individuals in the GA given the slightly more complex model. The fluorescence image that we fit the model to only stained for living cells and dead cells, and we decided to fit only to the locations of the dead cells for simplicity. In the imaging data, there was some overlap between living and dead cells that was not captured with the model. A more complex model, with more parameters, is needed to fully reproduce the behavior shown in the experimental images. However, to test our method and demonstrate its utility, we only used dead cells for fitting. A key difference that separates this test from the previous, besides the fact that we are now comparing model simulations to real data, is that the image is spatially much larger than the tumor that we are simulating. The diameter of the image is roughly 5,500 microns, while we use the ABM to simulate a tumor of approximately 3,000 microns in diameter (150 cells). In terms of cell numbers, simulating a tumor of the actual size would involve roughly 3.5 times more cells than the simulation that we performed. This would greatly increase the computational time needed to run each simulation and perform parameter estimation. Additionally, tumors taken from mouse models and patients are even larger, and, in most situations, it is prohibitive to simulate at a true biological scale. Therefore, while it would have been feasible for us to simulate a tumor that is the same size as this image, we deliberately simulate a smaller tumor to test how our approach functions when comparing across scales. Thus, we specifically aim to demonstrate that our approach works well even when the sizes of the tumors simulated by ABMs are different from the size of the tumor that was imaged.

Parameter fitting was performed in a similar manner as the first example. In addition to narrowing parameter bounds using the distance between tumor and simulated images based on the low-dimensional representation, we also adjusted some parameter bounds based on fitting results. We recognize that this may insert some bias into the fitting and could potentially be avoided by performing a larger number of initial simulations. However, our goal with this work is primarily to display the use of representation learning for calculating an objective function, while balancing the use of available computational resources.

The final fit is shown in **Fig 5A**. As in the first example, not many fitting steps ($< 10$) were required before the best and average fits leveled off. Visually comparing the processed image for the experimental data to the processed model simulation (**Fig 5B**), we see that the best-fit simulation bears strong resemblance to the tumor image, confirming the fitting result.

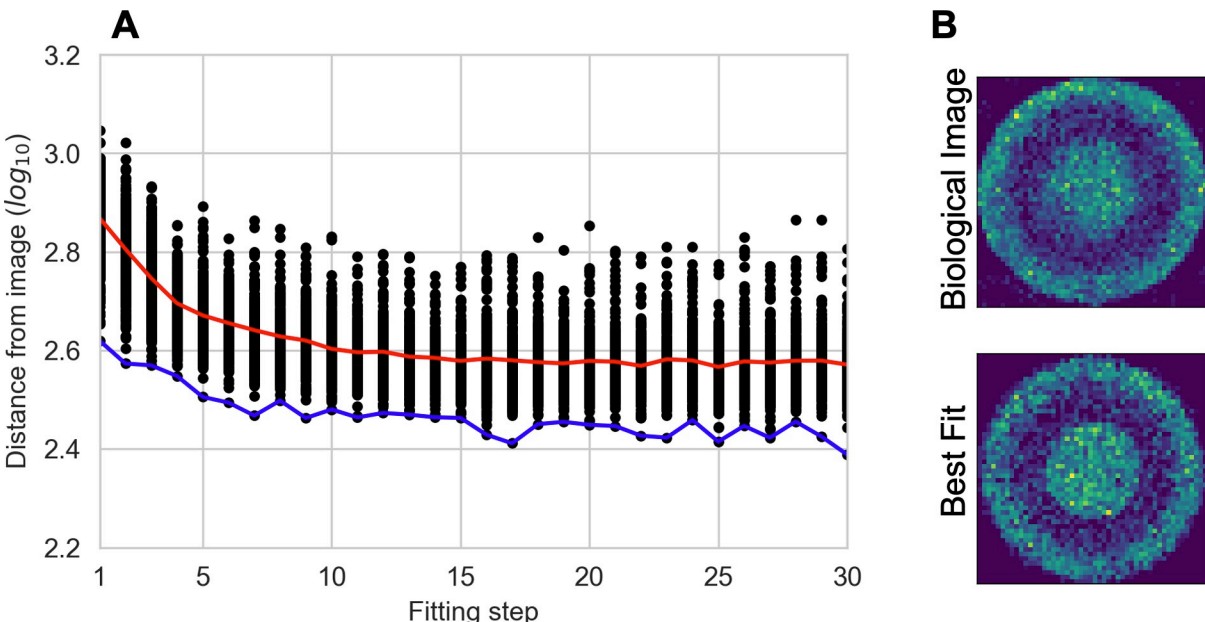

**Fig 5. Fitting results for Example 2.** (A) Fitting with a GA. Black dots–individuals in the GA. Red line–average fit. Blue line–best fit. (B) Visual comparison of the processed image for the tumor image (top) and the best-fit simulation (bottom).

Although the actual values of the fitted parameters are not the focus of this study, we demonstrate that best-fit parameters are tightly constrained and identifiable (**Table 2**). Overall, this example not only displays that our approach can be used to compare ABM simulations to real-world images, but that it can be used to fit parameters across spatial scales.

## Discussion

With this study, we present a method for performing a quantitative comparison between tumor images and ABM simulations and fitting model parameters. To our knowledge, this is the first method for truly performing such a comparison, accounting for spatial layouts without having to manually calculate differences between tumor images and model simulations for user-defined comparison metrics. We compare the images and model simulations using representation learning to project model simulations and tumor images into low-dimensional space and then calculate the distance between the two. Using this distance as an objective function, we are able to fit to both model-generated data and actual biological data. Importantly, we show that with the data-processing step, we are able to fit across spatial scales. This is vital for tumors simulated using ABMs, as they are generally much smaller than actual tumors.

**Table 2. Top 10 best fits from fitting an ABM to tumor imaging data for Example 2.**

|  | T cell killing probability | T cell infiltration | Cancer cell death probability | Radius of hypoxic area | Death probability in hypoxic area |
|---|---|---|---|---|---|
| **Best Fit** | $3.99 \times 10^{-3}$ | 0.386 | $9.98 \times 10^{-4}$ | 641 | $5.00 \times 10^{-3}$ |
| **Maximum** | $3.99 \times 10^{-3}$ | 0.398 | $9.99 \times 10^{-4}$ | 698 | $5.00 \times 10^{-3}$ |
| **Minimum** | $3.95 \times 10^{-3}$ | 0.382 | $9.86 \times 10^{-4}$ | 636 | $4.81 \times 10^{-3}$ |
| **Mean** | $3.98 \times 10^{-3}$ | 0.391 | $9.96 \times 10^{-4}$ | 647 | $4.94 \times 10^{-3}$ |
| **Coefficient of Variation** | $3.94 \times 10^{-3}$ | $1.33 \times 10^{-2}$ | $3.98 \times 10^{-3}$ | $2.98 \times 10^{-2}$ | $1.21 \times 10^{-2}$ |

We believe this method provides researchers a way to more accurately set tumor ABM parameters based on actual image data. This would allow for a stronger bridge between computational modeling and clinical and experimental research by ensuring that ABMs more accurately represent tumor images. Additionally, this could have broader impacts beyond ABMs used for mathematical oncology, as it can be applied to fit ABMs to spatially resolved data for a range of applications. Specifically, by modifying the data-processing step, this method could be applied to any type of spatial model that requires image data where an objective function would otherwise have to be designed based on specific metrics extracted from the image.

One aspect with our data-processing step that must be considered is the scaling of the simplified images. First, we note that this step can be ignored if the model is the same spatial scale as image data. However, when ABM simulations and data span different scales, two considerations are important. The first is that many parameters, such as diffusion rates, would have to be re-estimated due to the difference in spatial scale between the model and actual biology. In other studies, researchers scaled temporal parameters to account for this difference in scale [28], however other parameters may also be need to be re-estimated. This means that model parameters become specific to the difference in scale between the model simulations and the image used for parameter estimation, making it more difficult to transfer the parameters to a different model. The second consideration is that by cropping each simplified image to the tumor border, each simulation during parameter estimation is on a different scale. In this way, the scaling between the tumor image and the simulation becomes an additional parameter that is estimated.

Our method is of course not without limitations. The major limitation is that tumor images represent a single timepoint, meaning that our method is not temporally resolved. To improve this, an ABM should be fit simultaneously to an image along with temporal data. For example, the measured tumor volume over time can be used to further constrain parameter estimation. An additional limitation is related to the imaging data that can be acquired. If whole-tumor images are not available, the model simulations should be cropped to the same region as the image. Lastly, as with many parameter estimation approaches, this method is computationally expensive, as many simulations are needed in order to fit the neural network and to perform subsequent parameter estimation. There are strategies for potentially overcoming the computational resources required. In Example 2, we manually adjusted the bounds of some parameters and limited the size of the GA population. In addition, we note that the simulations required for initial estimation of the parameter ranges can be run in parallel, reducing the computational expensive. Finally, we acknowledge that our analyses of the parameter estimation results are limited. However, we chose not to include a critical evaluation of the estimated parameters and their biological implications in order to maintain focus of this study on implementation of representation learning as an objective function for tumor ABMs. Future work can provide an in-depth description of critical aspects of model calibration, including optimizing the cell types and interactions included, performing parameter sensitivity analysis, and analyzing the fitted parameter values.

## Conclusions

We developed a novel method for fitting ABM simulations to tumor images by using the distance between images and simulations in projected space as the objective function for parameter estimation. We show that this approach is successful, using both model-generated data and actual tumor images. Overall, our method provides a new way to determine ABM parameters beyond a visual comparison or simple user-defined metrics.

## Acknowledgments

The authors thank members of the Finley research group for critical feedback on the manuscript and Dr. Keyue Shen for making published data from their *in vitro* tumor model readily available.

## Author Contributions

**Conceptualization:** Colin G. Cess, Stacey D. Finley.

**Formal analysis:** Colin G. Cess.

**Investigation:** Colin G. Cess.

**Methodology:** Colin G. Cess.

**Resources:** Stacey D. Finley.

**Supervision:** Stacey D. Finley.

**Writing – original draft:** Colin G. Cess.

**Writing – review & editing:** Colin G. Cess, Stacey D. Finley.

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
