## [Decision Letter · Decision Letter 0]

21 Feb 2023

Dear Dr. Finley,

Thank you very much for submitting your manuscript "Fitting agent-based models to tumor images using representation learning" for consideration at PLOS Computational Biology. As with all papers reviewed by the journal, your manuscript was reviewed by members of the editorial board and by several independent reviewers. The reviewers appreciated the attention to an important topic. Based on the reviews, we are likely to accept this manuscript for publication, providing that you modify the manuscript according to the review recommendations.

Sincerely,

Pedro Mendes, PhD

Academic Editor

PLOS Computational Biology

Kiran Patil

Section Editor

PLOS Computational Biology

Reviewer's Responses to Questions

**Comments to the Authors:**

Reviewer #1: I enjoyed reading this manuscript. I think this tool will be useful for the challenging task of calibrating agent-based models. Some suggestions for improving the presentation and clarifying needed details:

The central question I was left with after reading the manuscript is: ‘What ‘low’ dimension is used here? There may be some discussion of this in your earlier contribution on the learning network architecture, but it needs to be discussed here as well. Even the order of magnitude is a bit of a mystery: 3? 30? 300? Perhaps some guidelines/strategy for selecting this dimension (and any constraints on that choice) could be discussed in the introduction and then the specific choices made in examples 1 and 2 could be provided and justified.

On a related note, it would be useful to compare your representation learning results to more standard dimensionality-reduction methods (e.g. PCA), for instance, in terms of the dimension of the reduced description. One would not expect PCA to capture the nonlinear relationships between parameter values and ABM simulation output, but this comparison might help your readers better appreciate your contribution.

Training: It would be good to expand on how to approach training in a more ‘realistic’ scenario where there is an ensemble of experimental images available. It seems that this that case it would be advisable to train on images as well as simulations. Can you speak to any challenges or opportunities that might present?

More specifically, for the training described in the manuscript: Are the initial conditions of the simulations widely sampled like the parameter values, or are they fixed to be consistent with the conditions of the experiments? How may training simulations were generated?

Minor points:

Intro: may want to also highlight the fact that (most) ABM formulations are stochastic, which further complicates calibration.

Line 25: it’s implied that ‘time-course data’ is non-spatial, which is certainly most commonly the case, but time-lapse microscopy data isn’t captured here. Perhaps add the modifier ‘aggregate’ to clarify.

Line 93: “To this end, we endeavored to develop a method for quantitatively comparing experimental tumor images to ABM simulations as a single value.” The phrase ‘as a single value’ is hard to interpret here. Maybe this could be reworded something like ‘To this end, we endeavored to develop a method for establishing a scalar measure for comparison of experimental tumor images and ABM simulations.”

Line 166 “The second is that it converts discrete cell locations into regions of cell density, allowing us to compare across spatial scales.” You’ve used the verbs ‘convert’ and ‘reduce’ in this description, but it may be useful to use the term ‘aggregate’ to better convey the meaning. Also, there’s some ambiguity in using ‘grid’ to refer to the lattice and to lattice-points. E.g. in Fig 2 caption ‘grids are then reduced to a smaller size’ maybe could be reworded something like: ‘grid-points are then aggregated, resulting in a coarser grid, where values correspond to local densities.’ Finally, Fig 2B: The figure doesn’t suggest that the size has changed (as in A), but rather a coarser grid has been applied (so the image is smaller in terms of pixel dimensions, but not inches on the page).

Line 179: “Following image processing, the next step of our method is to generate the model simulations needed to train the neural network.” This wording may be misleading as could be understood to suggest that images are processed before they are generated. Perhaps you mean to say ““Following the establishment of an image processing protocol, the next step of our method is to generate the model simulations needed to train the neural network (to which that image processing protocol will be applied).

Line 209: perhaps reword: “We then take the n closest simulations to the image and examine their parameter *values*

Caption fig 3: The wording here suggests that the choice of ‘nearby’ parameter value sets is based on a distance threshold, whereas in the text it’s indicated that the nearest ‘n’ parameter value sets are selected. Either are reasonable, but the wording can be made consistent.

Line 219: ABM description. You’ve made clear that this simulation is being run just for illustration, but it’s still a bit concerning not to have a full description of the ABM details. Perhaps the code could be provided? That would ensure that a reader would be in a position to replicate your results.

Line 277: best to describe this as the endpoint of the simulation, yes?

Line 280: “the based parameters” -> ‘base parameters’, ? You may consider using the more specific term ‘nominal’ instead of ‘base’ for your reference simulation. Likewise in table 1 the ‘base parameter value’ is inconsistently referred to as ‘actual’

Table 1: the line between maximum and mean is oddly placed. Not clear why it’s there. Also, it might be more useful to report the CoV instead of s.d., especially here where the number format is not consistent. Finally, it would be interesting to see the max/min search range that you’re exploring (to get an idea of how narrow the search space is, and whether any of the best fits are worryingly close to those bounds). This is especially important in example 2 where the manual choice of parameter search region muddies the description of the search.

Line 295: “of approximately 3,000 microns in diameter or 150 cells” Ambiguous wording, I presume this means “of approximately 3,000 microns (i.e. 150 cells) in diameter”

Example 2: the ‘across scales’ fitting argument is a little hard to follow. As the ABM designer, you can select how coarse to set your grid, so you could interpret your model as clusters of cells at each grid-point. Is that the interpretation being used here?

Line 302: manual setting of bounds is fine, but is inconsistent with your previous description of the algorithm (bounds set in terms of distance in the low-dimensional representation). Were these both used? Or the latter not employed?

Line 331: “We compare the images and model simulations using representation learning to project model simulations and tumor images into low-dimensional space and then calculating the distance between the two.” Grammar: ‘and then calculate…’

Reviewer #2: This is a very interesting and well-done paper on varying model/cell-behavior parameters and adjusting them such that the output is visually/spatially realistic. All of these comments below are relatively minor.

I would suggest changing the title from “fitting” to “validating” or “calibrating and validating.” In my experience, non-ABM people often use the term fitting pejoratively. I leave it up to you, this has just been my experience in the field.

It would be useful to cite other examples of GA’s being used for parameter exploration in ABMs.

On line 260, you state, “The model is stochastic, so we cannot expect to fit the parameters perfectly.” Is there a range of parameters that generate equivalent results over some number of stochastic replicates? Or is the supposition that there is a unique best fit?

On line 299, the authors state “and it is currently impossible to simulate tumors on that scale.” This is debatable are there are biomedical ABMs out there that operate at anatomic scale and incorporate tens of billions of cells.

If there are any plans for extending this to 3d tumors, that should be included in the discussion.

As a matter of style, superscript citations typically come after periods. There are a few minor typos also (line 370: “computational expensive.”)

Reviewer #3: Uploaded as attachment.

**Have the authors made all data and (if applicable) computational code underlying the findings in their manuscript fully available?**

Reviewer #1: Yes

Reviewer #2: Yes

Reviewer #3: Yes

PLOS authors have the option to publish the peer review history of their article (what does this mean?). If published, this will include your full peer review and any attached files.

Reviewer #1: No

Reviewer #2: No

Reviewer #3: No

Figure Files:

Data Requirements:

Reproducibility:

References:

---

## [Editor Report · Decision Letter 1]

3 Apr 2023

Dear Dr. Finley,

We are pleased to inform you that your manuscript 'Calibrating agent-based models to tumor images using representation learning' has been provisionally accepted for publication in PLOS Computational Biology.

Best regards,

Pedro Mendes, PhD

Academic Editor

PLOS Computational Biology

Kiran Patil

Section Editor

PLOS Computational Biology

---

## [Editor Report · Acceptance letter]

19 Apr 2023

PCOMPBIOL-D-23-00146R1 

Calibrating agent-based models to tumor images using representation learning

Dear Dr Finley,

I am pleased to inform you that your manuscript has been formally accepted for publication in PLOS Computational Biology. Your manuscript is now with our production department and you will be notified of the publication date in due course.

With kind regards,

Zsofi Zombor
